# GAIN: A Gated Adaptive Feature Interaction Network for Click-Through Rate Prediction

**DOI:** 10.3390/s22197280

**Published:** 2022-09-26

**Authors:** Yaoxun Liu, Liangli Ma, Muyuan Wang

**Affiliations:** College of Electronic Engineering, Naval University of Engineering, Wuhan 430033, China

**Keywords:** recommendation system, factorization machines, feature interaction, deep learning

## Abstract

CTR (Click-Through Rate) prediction has attracted more and more attention from academia and industry for its significant contribution to revenue. In the last decade, learning feature interactions have become a mainstream research direction, and dozens of feature interaction-based models have been proposed for the CTR prediction task. The most common approach for existing models is to enumerate all possible feature interactions or to learn higher-order feature interactions by designing complex models. However, a simple enumeration will introduce meaningless and harmful interactions, and a complex model structure will bring a higher complexity. In this work, we propose a lightweight, yet effective model called the Gated Adaptive feature Interaction Network (GAIN). We devise a novel cross module to drop meaningless feature interactions and preserve informative ones. Our cross module consists of multiple gated units, each of which can independently learn an arbitrary-order feature interaction. We combine the cross module with a deep module into GAIN and conduct comparative experiments with state-of-the-art models on two public datasets to verify its validity. Our experimental results show that GAIN can achieve a comparable or even better performance compared to its competitors. Furthermore, in order to verify the effectiveness of the feature interactions learned by GAIN, we transfer learned interactions to other models, such as Logistic Regression (LR) and Factorization Machines (FM), and find out that their performance can be significantly improved.

## 1. Introduction

With the explosion of mobile internet data, almost all Information Technology (IT) enterprises have gathered massive data, such as users’ personal information and their interactive information with the user interface, using their own Apps. These data can be used to learn interests or consumption habits of various users, so as to provide them with personalized recommendations, which has become one of the most critical profit engines. For example, content sharing platforms, such as YouTube and TikTok, increase user stickiness by pushing them videos which they may be interested in, and e-commerce platforms, such as Amazon and Taobao, facilitate transactions through the individual customized recommendations of commodities.

The Click-Through Rate, which is a critical indicator in a recommendation system, determines what content should be displayed and in what order. For those IT enterprises, even a slight improvement of 0.001 can be considered as practically significant [1,2] and will bring billions of dollars in extra revenue. Therefore, the CTR prediction task has always been a hot research spot for academia and industry.

Modeling a feature interaction is one of the most prevailing methods for the CTR prediction task. In an early study [3], feature interaction is also called cross feature, and a combination of different features is regarded as a new feature. However, meaningful cross features rely on the manual engineering of domain experts, which is usually a very time-consuming job. Although FM and its variants [4,5,6] make it possible to learn $2_{nd}$-order interactions automatically, learning higher-order interactions remains impractical due to its high computational complexity. Inspired by the triumph of deep learning in computer vision [7,8], speech recognition [9,10] and natural language processing [11,12], many researchers resort to deep neural networks (DNNs) to learn high-order feature interactions and obtain decent results [13,14]. Nevertheless, some studies [15,16,17] point out that DNNs model feature interactions in an implicit way and can hardly learn interactions higher than the $2_{nd}$ order. In recent years, integrated models which combine a deep module with a shallow module (here, we use the word “shallow” to distinguish from “deep”) have become the most prevailing solution.

The models that emerged in recent years can be seen as improvements to the above works and can be roughly divided into three categories. Models from the first category try to improve the model performance by ameliorating feature embeddings. All features should be embedded to vectors before interacting with others, and in typical CTR models, each feature usually corresponds to one embedding vector. Juan et al. [6] introduce the concept of a “feature field” and assign multiple vectors to each feature, so a feature can use different embedding vectors to interact with other features from different fields. Yang et al. [18] argue that each feature should use different representations when performing different operations and propose a new embedding method. Assigning multiple embedding vectors to one feature creates too many parameters, and thus affects the model’s efficiency. To solve this problem, Sun et al. [19] propose to use a field matrix between two feature vectors to model their interactions.

The models that form the second category are committed to the exploration of the interaction form. The inner product, outer product and Hadamard product are commonly used interaction operations in CTR models. Some works [20,21,22] state that these operations are too simple to effectively model feature interactions and propose their own interaction operations to obtain a better expressive effect. A self-attention mechanism [11], which can be regarded as an interaction operation, has been utilized in CTR prediction models [23,24,25]. In addition, Lang et al. [26] design a model to adaptively explore and optimize the interaction operations according to data.

Models from the third category work on ameliorating model architectures. Integrated models usually consist of a shallow module and deep module. Some models [5,20,27,28,29] use a single-tower architecture, in which two modules work in sequence, while other models [3,17,30,31,32] choose a dual-tower architecture, in which two modules work in parallel. Chen et al. [33] point out that insufficient sharing between two modules may limit the model’s expressiveness and effectiveness and devise a bridge module to enhance the layerwise information sharing. Song et al. [34] introduce a neural architecture search (NAS) to CTR models and utilize the learning-to-rank strategy to search the optimal architecture.

Zhu et al. [1] design an open benchmarking framework for CTR prediction models, which provides a fair comparison environment to evaluate the performance of different models. After fine-tuning and a thorough comparison, several interesting observations are worth noting. Firstly, in spite of the diversity of architectures and interaction manners, the differences in model performance are smaller than expected. Secondly, DNNs still serve as a strong baseline, and a plain multi-layer perception (MLP) performs better than complicated deep models [35,36]. Thirdly, the complex design of the architecture decreases the model’s efficiency and does not always lead to an improvement in the model performance. How to make a trade-off between complexity and efficiency is indeed a very challenging task.

In this paper, we propose a lightweight, yet efficient feature interaction model, whose architecture is shown in Figure 1. As the core of our model, the cross (shallow) module is composed of multiple interaction units, which learn feature interactions independently. Each unit preserves a gate for every feature field to decide whether the field will engage in an interaction. Each gate has only two statuses, closed or open, which are randomly initialized and adaptively updated according to the data. For each unit, the order is determined by the number of fields involved in the interaction, i.e., the number of open gates. Therefore, each unit can adaptively learn an arbitrary-order interaction without assigning an order previously. Theoretically, the learning ability of our cross module is limited by the number of units. However, the experiments demonstrate that the model performance does not always upgrade with the number of units, and excessive units may bring down the prediction accuracy. Instead of considering all possible interactions, our cross module aims to learn useful ones which only take up a small proportion. In addition, our proposed model includes a deep module which is implemented by an MLP. The two modules work in parallel and share the same feature embeddings. A cross module can memorize frequent combinations which are discriminative to the prediction, while a deep module can generalize an interaction to unseen combinations. We name our model the Gated Adaptive feature Interaction Network, abbreviated GAIN. To summarize, we make the following contributions:1.We devise a novel cross module to learn interactions. The structure of our cross module is quite simple; in contrast to shallow modules in other models which consist of multi-layer or complex connections, our cross module is composed of multiple gated units. Each unit works independently and can take full advantage of GPU parallel computing, which ensures the efficiency of our model.2.We propose three strategies to learn useful feature interactions. Three strategies have their own characteristics and are suitable for different scenarios.3.By assembling the cross module and deep module, we propose the Gated Adaptive feature Interaction Network (GAIN) to adaptively learn useful feature interactions.4.We carry out extensive experiments on two real-world datasets to validate the effectiveness and efficiency of GAIN.

**Figure 1 sensors-22-07280-f001:**
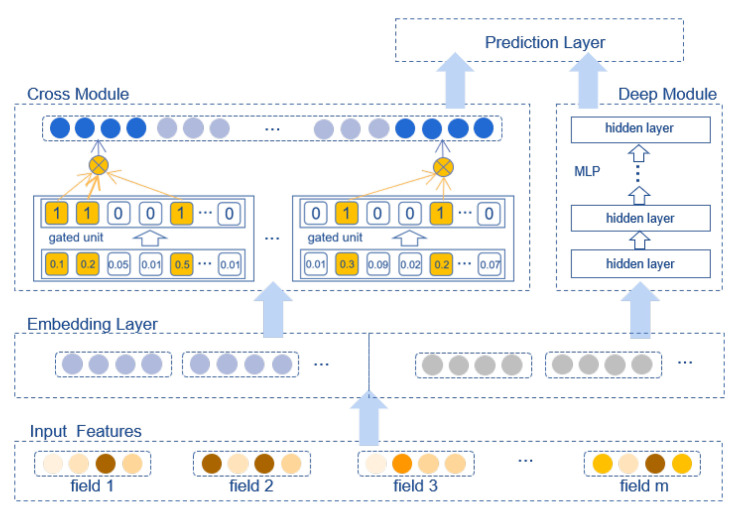
Architecture of GAIN.

The rest of this paper is organized as follows. Section 2 introduces the related works which are relevant to our proposed model which is described in detail in Section 3. We analyze our model from different aspects in Section 4. The experimental results on three real-world datasets are presented and discussed in Section 5, followed by Section 6, which provides a conclusion of our work.

## 2. Related Works

### 2.1. Feature Embedding

Real-world CTR datasets usually consist of multiple feature fields, and each field contains multiple features. In order to make full use of the information in the features, raw features should be transformed to expressive representations. In early studies, a common approach is to represent every feature as a one-hot vector. However, one-hot encoding brings sparsity to feature vectors, especially when a feature field contains a large number of features, which would result in an excessive memory assumption and a decrease in model efficiency. In fact, in a dataset with tens of millions of samples, it is very common for a feature field to contain millions of features. As a result, one-hot encoding is no longer applicable for a nowadays web-scale dataset. Currently, the mainstream approach is feature embedding, i.e., encoding features to low-dimensional dense vectors. Categorical features and numerical features are usually processed separately.

#### 2.1.1. Categorical Feature

For a categorical feature set $\left[F_{1},F_{2},\cdots ,F_{i},\cdots ,F_{m^{{}^{\prime }}}\right]$, $m^{{}^{\prime }}$ denotes the number of categorical feature fields, and the $i_{th}$ feature field $F_{i}$ contains $n_{i}$ different features. A vocabulary consists of $n_{i}$ vectors and is built for $F_{i}$, then every feature can correspond to a unique dense vector.

In vector-wise feature interaction models [27,31,32,37,38], every feature interacts with others as a whole; hence, all features need to be embedded to dense vectors with the same length. Other bit-wise feature interaction models [17,39] argue that it is irrational to embed all features into the same length because the numbers of features in different fields vary widely and concatenate all feature embedding vectors to a one-dimension vector. Therefore, in these bit-wise models, the length of the embedding vector for different fields can be arbitrary.

In most models, each feature usually corresponds to only one vector, while in some models, such as Field-aware Factorization Machines (FFM) [6] and Operation-aware Neural Networks (ONN) [18], in order to enhance the model’s expressiveness, each feature corresponds to multiple vectors; hence, different embedding vectors can be used to interact with features from different fields. In Field-matrixed Factorization Machines (FmFM) [19], though each feature corresponds to one vector, a field matrix is proposed to interact with different field pairs, which enhance the expressiveness of the model while maintaining computational space and time efficiency.

#### 2.1.2. Numerical Feature

For a numerical feature set $\left[F_{1},F_{2},\cdots ,F_{i},\cdots ,F_{m^{{}^{\prime \prime }}}\right]$, $m^{{}^{\prime \prime }}$ denotes the number of numerical feature fields. The value range of feature field $F_{i}$ contains innumerable continuous values which make it impossible to encode the numerical features to dense vectors such as categorical features. The most widely used approach in CTR models is discretization, which aims to discretize numerical features to categorical ones. The value range of $F_{i}$ can be partitioned into multiple segments denoted as $\left[Seg_{1},Seg_{2},\cdots ,Seg_{i},\cdots ,Seg_{n_{i}}\right]$, where $Seg_{i}$ denotes the $i_{th}$ segment, and $n_{i}$ denotes the number of segments of field $F_{i}$. Each segment corresponds to a unique embedding vector. For $Seg_{i}$, all values in it share the same embedding vector $v_{i}^{num}$. However, this approach relies too much on partitioning strategies and can easily lead to local optima.

In models such as AutoInt [23], the Adaptive Factorization Network (AFN+) [32] and ContextNet [27], every numerical feature field is assigned a vector, and a feature embedding is represented with the production of its value and the vector of the field it belongs to. For example, the vector set of all numerical feature fields is denoted as $\left[v_{1}^{num},v_{2}^{num},\cdots ,v_{i}^{num},\cdots ,v_{m^{{}^{\prime \prime }}}^{num}\right]$, and for an arbitrary value α of the $i_{th}$ feature, its embedding vector can be represented as $\alpha \cdot v_{i}^{num}$.

AutoDis [40] initializes multiple meta-embeddings for every numerical field and assigns every feature with a unique vector through the weight-sum operation of multiple meta-embeddings. The deep learning recommendation model (DLRM) [28] concatenates all numerical features in a sample and transforms it to a vector with equal size to a categorical feature through an MLP.

Instead of encoding numerical features to a vector, some models maintain the original form of a scalar for every feature, e.g., Wide & Deep [3] uses original values, the Deep & Cross Network (DCN) [17] and DCNv2 [39] normalize all values of each field and the DNN for YouTube [14] utilizes several transformations of normalized values.

### 2.2. High-Order Feature Interaction

The Factorization Machine (FM) [4] assumes that every feature can be represented with a low-dimensional embedding vector; hence, the $2_{nd}$-order feature interaction can be implemented by using the inner product of two vectors. Although the FM can learn the feature interaction automatically, it can hardly learn an interaction higher than the $2_{nd}$ order because a large number of meaningless feature pairs involved in the computation procedure would raise the complexity.

It has been a consensus that learning high-order features interaction is critical for CTR prediction. Models such as Higher-Order Factorization Machines (HOFM) [41], DCN, xDeepFM [31], DCN-v2 and the Field-aware INTeraction Neural Network (FINT) [37] are proposed to model high-order feature interactions by stacking multiple cross layers. These models share three main drawbacks. Firstly, the model capacity is restricted by the number of cross layers which determines the maximum order. Secondly, excessive interactions brought by computation will degrade the model performance. Thirdly, in a stacked structure, feature interactions have to be calculated layer by layer, which will bring down the efficiency.

The Deep Interaction Machine (DeepIM) [38] introduces Newton’s identities and represents an arbitrary-order interaction in the form of an elementary symmetric polynomial, which can be calculated by the power sums of features embedding vectors. Using this trick, the original complex computation of multiple layers is simplified to an algebra problem. However, the trick is a two-edge sword; the symmetry of Newton’s identities makes it hard to differentiate the importance of interactions, which would limit the model’s expressiveness.

### 2.3. Selection of Useful Feature Interaction

The computation process of those models described in Section 2.2 involves all the possible interactions of bounded degree without distinguishing their importance, which would significantly increase the complexity and harm the performance. With the rise of the attention mechanism, many attention-based models are proposed to cope with this problem. Attentional Factorization Machines (AFM) [42] utilize the attention mechanism to enable interactions to contribute differently to the prediction. To enhance the model’s expressiveness, AutoInt [23] introduces a multi-head self-attention mechanism, which is updated to disentangled self-attention in Disentangled Self-Attentive Neural Networks (DESTINE) [24]. However, attention calculates the similarity of two vectors, and it cannot be used to represent the importance of the feature interaction.

Xue et al. [43] introduce the hashing algorithm to the feature interaction and propose AutoHash, whose core idea is to put *m* features into *k* buckets according to a hash function, each feature can be reused and the feature combination in each bucket represents a feature interaction. The hash function is determined by probabilistic distribution *p*, which is a learnable parameter. The authors believe that AutoHash can adaptively learn useful feature interactions in a data-driven way.

An automatic feature interaction selection (AutoFIS) [44] proposes a two-stage algorithm and assigns a gate for every possible interaction to control whether the interaction would be calculated in the next step. In the search stage, the importance of every interaction is represented as an architecture parameter α that can be optimized by gradient descent. Those interactions who’s corresponding α equals to zero would be regarded as useless and then be dropped. In the re-train stage, the α of the remaining interactions will serve as attention units and the model will be trained for the second time.

Chen et al. [45] propose a Bayesian higher-order feature interaction selection (BH-FIS) to perform the FIS. First, the outer product and masking techniques are utilized to enumerate all feature interactions. Second, variational inference networks are leveraged to calculate coefficients for all feature interactions. AFN+ [32] represents a feature interaction in the form of a power product of features and projects embedding vectors into a logarithmic space. Logarithmic neurons are devised to learn useful interactions. Each neuron is independent from others and can learn arbitrary-order interactions adaptively. Although logarithmic transformation can convert the operation of a power product to a weighted sum, it imposes constraints on the values of input because the log function requires positive inputs. AFN+ adopts the method of taking the absolute value of the original input, which not only might lead to a collision but also influence the semantic of embedding vectors.

Our proposed model GAIN is inspired by AutoFIS and AFN+. We assume that only a minority of useful interactions would bring about appreciable influence to the prediction accuracy, and these useful interactions can be learned through training. Unlike AutoFIS, GAIN accomplishes searching and training in one stage. There is no need for it to consider all the possible interactions as useful ones can be learned automatically. Compared with AFN+, GAIN restricts the value of the power exponent of the embedding vectors to 0 or 1 and relaxes the restriction on the value of the inputs.

## 3. Gated Adaptive Feature Interaction Network

GAIN applies the classic structure in CTR prediction task which consists of embedding layer, cross module, deep module and prediction layer. Cross module and deep module work in parallel and share the same inputs from embedding layer. Outputs of the two modules are passed to the prediction layer. In this section, we will describe these components in detail.

### 3.1. Embedding Layer

GAIN assembles two fashions of feature interaction. The cross module models explicit interaction at vector-wise level, which requires that all features must be embedded to vectors with same length. The deep module models implicit interaction at bit-wise level, which has no requirements on the length of embedding vectors. Some studies [6,18] illustrate that using different representations for different operations can lead to better performance. However, parameters in embedding layer account for a high proportion of total parameters in the model, and using different embedding vectors in two modules will lead to a significant increase in model’s complexity.

To facilitate interaction with others, all features, including numerical and categorical features, are encoded to dense vectors with equal size. First, numerical features are converted to categorical features through discretization. After that, a vocabulary is randomly initialized for each field, and every feature is represented with a unique vector by means of a look-up table. At last, all feature embedding vectors are assembled to form the input of cross module and deep module, respectively. The input of cross module is represented as a matrix $X_{cross}=V\in R^{d\times m}$, where *d* and *m* represent the dimension of embedding vectors and the number of feature fields, respectively. The input of deep module is represented as a one-dimension vector $X_{deep}=Cat\left[v_{1},\cdots ,v_{i},\cdots ,v_{m}\right]$, where $v_{i}$ is embedding vector of the $i_{th}$ feature.

Uniqueness is the basic requirement of feature embedding; besides that, every embedding vector should carry certain information, i.e., features co-occurring frequently in positive samples should show some similarity in latent representation space. Although randomly initialized vectors can ensure uniqueness, they cannot carry useful information which is deeply buried in data. GAIN can be trained end to end, and the embedding layer is jointly optimized with other components, so as all embedding vectors can be updated according to data.

### 3.2. Cross Network

The cross module is composed of multiple gated units, each of which can adaptively learn a unique feature interaction. Each unit consists of *n* gates, and each gate controls whether the corresponding feature engages in the interaction. Every single feature is $1_{st}$ order, and the combination of different features is regarded as the feature interaction and the number of features in the combinations is the order of interaction. A generalized definition of feature interaction is given as below.

**Definition** **1.**
*Let $F=[F_{1},\cdots ,F_{i},\cdots ,F_{m}]$ denote the collection of feature fields, the combination of k features $F_{t_{1}}=\alpha ,F_{t_{2}}=\beta ,\cdots ,F_{t_{k}}=k$ is a $k_{th}$-order feature interaction, where $1\leq t_{1},t_{2},\cdots ,t_{k}\leq m$, and $F_{i}$ represents the $i_{th}$ field.*


To facilitate the multiplication operation, a definition based on embedding vectors is given as below.

**Definition** **2.**
*Let $V^{d\times m}$ denote the embedding matrix of an instance in datasets, and $v_{i}$ denotes the $i_{th}$ row of $V^{d\times m}$. Then, a $k_{th}$-order feature interaction can be represented as*

(1)
$${\mathrm{FI}}^{\left(k\right)}=\prod_{i=1}^{m}v_{i}^{g_{i}}\;$$

*where $k=\sum_{i=1}^{m}g_{i}$ is the interaction order, and $g_{i}\in \left\{0,1\right\}$ is the gate status.*


#### 3.2.1. Gated Unit

A gating mechanism is introduced into the cross module, each unit maintains *m* gates and each gate controls the engagement of one feature field. There are only two statuses for gate, closed or open, which indicates the corresponding feature will be dropped or reserved, respectively. Combination of *m* gates in a unit can be regarded as a feature interaction. Each unit can learn a unique feature interaction, and expressiveness of our proposed model is determined by the number of units *n*. However, our experimental results demonstrate that model performance does not always improve with the increase in *n*; therefore, finding an appropriate *n* is crucial to GAIN.

The statuses of gates are initialized randomly and updated dynamically by gradient decent. For convenience of calculation, the statuses of gates in a unit are represented as a multi-hot vector, in which open and closed gates are denoted as 1 and 0, respectively. However, this manner of representation cannot be directly used in training phase because discrete statuses cannot propagate gradients. To overcome this problem, a continuous relaxation technology is introduced to the status of gates. Firstly, we randomly initialize a weight vector for each unit, then a softmax function is utilized to transform the weight vector to a probability distribution, which is called “soft” vector (by contrast, the multi-hot vector is called “hard” vector). Secondly, “soft” vector is transformed to “hard” vector with specific feature selection strategy (which will be described in Section 3.2.2). In “hard” vector, a value of 1 indicates that its corresponding feature will engage in interaction. In the forward phase of training, “hard” vector is used to determine which features will be selected, while in the backward phase, gradients will not be backpropagated to “hard” vector but “soft” vector. In particular, given an initialized weight vector [0.8,0.6,0.5,0.1,0.2], the “soft” vector [0.28,0.23,0.20,0.14,0.15] can be easily calculated. By applying a threshold strategy which selects features whose probabilities greater than 0.20, we obtain the “hard” vector [1,1,1,0,0], which is passed forward to calculate feature interaction. In the backward phase, the gradients are backpropagated to the “soft” vector and the weight vector in sequence, skipping the “hard” vector. Assuming that the weight vector is updated to [0.9,0.5,0.2,0.1,0.5], the “soft” vector for next minibatch becomes [0.31,0.20,0.15,0.14,0.20], and the “hard” vector becomes [1,1,0,0,1]. During training, the weights are constantly updated and eventually stabilized, which will help the units to learn meaningful feature interactions. Those features which are more discriminative to prediction will accumulate more gradients and will be more likely to be selected. The data flow of a gated unit is shown in Figure 2.

In a gated unit, engagements of features are determined by “soft” vector, but only those with higher probabilities will have the chance to engage in interaction, while features with lower probabilities will lose their chance to be selected, which will not only make the “soft” vector lose the meaning of probability but also lead to local optima. Gumbel-softmax [46], a reparameterization trick, is introduced to cope with this problem. Gumbel-softmax trick is implemented with two steps.

Firstly, random numbers from a Gumbel distribution are added to every element of “soft” vector. The “soft” vector can be represented as
$$S=[s_{1},\cdots ,s_{i},\cdots ,s_{m}].$$

The random numbers from a Gumbel distribution can be represented as
$$\varphi_{i}=-log\left(-log\left(\mu_{i}\right)\right),$$
where $\mu_{i}\in U\left(0,1\right)$ is uniformly distributed random numbers. Then, the new “soft” vector can be represented as
(2)$$S^{{}^{\prime }}=\left[s_{1}+\lambda \varphi_{1},\cdots ,s_{i}+\lambda \varphi_{i},\cdots ,s_{m}+\lambda \varphi_{m}\right],$$
where λ is the coefficient of the Gumbel random numbers, which will be discussed in Section 5.3. Random numbers make all features have opportunities to engage in interaction and can help GAIN escape from local optima.

Secondly, a softmax function is leveraged to transform the new “soft” vector to a probability distribution which can be represented as
(3)$$S=[p_{1},p_{2},\cdots ,p_{i},\cdots ,p_{m}],$$
where $p_{i}=\frac{e^{S_{i}^{{}^{\prime }}/\tau }}{\sum_{j=1}^{m}e^{S_{i}^{{}^{\prime }}/\tau }}$, and τ is the temperature parameter, when τ→0, $p_{i}$ becomes a one-hot vector, and when τ→1, $p_{i}$ becomes a uniform distribution.

The significance of the Gumbel-softmax trick is that it not only retains the meaning of probability, that is, features with greater probabilities have greater probabilities of engagement in the interaction, but also makes the features with small probabilities have small chances to stand out. Note that the Gumbel-softmax function is replaced with a softmax function in the inference phase, which means that features are selected based on the probability distribution generated by the latest weight vector.

#### 3.2.2. Feature Interaction Selection

We design three strategies to select features from each gated unit. The first one is top-*k* strategy, which requires the order *k* to be specified in advance for each unit. In each unit, features with largest *k* probabilities will be selected. For convenience of representation, the order of all units can be specified in batches, e.g., dictionary {“3”:10,“4”:20,“5”:10} defines an allocation scheme of *k*, in which the keys represent the order and the values represent the number of units. With top-*k* strategy, we can manipulate our model to learn arbitrary high-order feature interactions as our wish. The process of FIS using top-*k* strategy in a gated unit is described in Algorithm 1.
**Algorithm 1:** FIS using top-*k* strategy.**Input**: Dictionary of orders: D.**Output**: Hard vector of gated unit: H.W← Randomly initialize weights of features;S← Compute Gumbel-softmax of W;k← Get order of unit *u* from D;$index^{\left(k\right)}\leftarrow$ indexes of the top-*k* probabilities in S;H← Generate multi-hot vector based on $index^{\left(k\right)}$.

   The second one is threshold strategy, which requires a global threshold ε. In each unit, features with probabilities greater than ε will be selected. The order of a unit is determined by the number of selected features, and may be different from the order of others. The orders of all units follow a Gumbel distribution which is relevant to the value of ε, and the effect of the ε on the distribution of orders will be discussed in Section 5.3. With threshold strategy, we do not need to assign orders in advance, gated units can adaptively learn feature interactions and the orders can be arbitrary. The process of FIS using top-*k* strategy in a gated unit is described in Algorithm 2.
**Algorithm 2:** FIS using threshold strategy.**Input**: Number of feature fields: *m*; Threshold: ε.**Output**: Hard vector of gated unit: H.W← Randomly initialize weights of features;S← Compute Gumbel-softmax of W;**for** *i=1 to m***do**
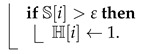


   The third one is hybrid strategy, which requires both threshold and allocation scheme of orders. Top-*k* strategy and threshold strategy are applied successively. Take a unit for an example. Firstly, largest *k* probabilities are coarsely selected. Secondly, probabilities less than threshold ε are filtered out. With hybrid strategy, our proposed model can not only reduce the impact of less important features on model performance but also control the orders within a reasonable range. The process of FIS using hybrid strategy in a gated unit is described in Algorithm 3.
**Algorithm 3:** FIS using hybrid strategy.**Input**: Number of feature fields: *m*; Threshold: ε; Dictionary of orders: D.**Output**: Hard vector of gated unit: H.W← Randomly initialize weights of features;S← Compute Gumbel-softmax of W;k← Get order of unit *u* from D;;$index^{\left(k\right)}\leftarrow$ indexes of the top-*k* probabilities in S;**for** *i in $index^{\left(k\right)}$***do**
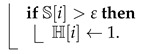


All of the three strategies can be used to adaptively learn useful feature interactions and have different application scenarios. Performance of the top-*k* strategy relies heavily on elaborate figuration of allocation scheme of *k*, which makes it more suitable for specific datasets with some prior knowledge. Threshold strategy which just requires a threshold can be easily applied on unfamiliar or desensitization datasets. Hybrid strategy combines the strengths of the two former strategies and are suitable for a variety of different scenarios.

#### 3.2.3. Multiplication Operation for Feature Interaction

Commonly used multiplication operations for feature interaction include inner product, outer product and Hadamard product. Inner product is widely used for its simplification and efficiency; meanwhile, it is criticized for its expressiveness as the result turns out to be a scalar. Outer product, which turns out to be a matrix, although boosts the expressiveness, brings huge computational complexity. Hadamard product is used in our model, because its result is still a vector, and it can be regarded as a trade-off between expressiveness and efficiency.

Most existing models depict the feature multiplication as a binary operation, hence learning *k*-order interaction needs to conduct multiplication operation for k−1 times. This may explain why those models stack multiple layers to learn high-order interactions. In order to devise a lightweight but efficient model, it is necessary to generalize the binary operation to a multivariate operation. The definition of multivariate operation of Hadamard product is given as below.

**Definition** **3.**
*Let $V^{{}^{\prime }}=\left[V_{t_{l}},V_{t_{2}},\cdots ,V_{t_{i}},\cdots ,V_{t_{k}}\right]$ denote the k features selected by a gated unit, and $V_{t_{i}}=\left[v_{i,1},v_{i,2},\cdots ,v_{i,d}\right]$ denotes the $i_{th}$ feature, then Hadamard product of these k features can be represented as*

(4)
$${\mathrm{HP}}^{\left(k\right)}=V_{t_{l}}⊗V_{t_{2}}⊗\cdots ⊗V_{t_{k}}=\left[\prod_{i=1}^{k}v_{i,1},\prod_{i=1}^{k}v_{i,2},\cdots ,\prod_{i=1}^{k}v_{i,d}\right],\;$$

*where ⊗ represents Hadamard product, k and d represent the interaction order and embedding dimension, respectively.*


${\mathrm{HP}}^{\left(k\right)}$, the output of a gated unit, remains a *d*-dimension vector. All units’ outputs are concatenated to form a one-dimension vector. Subsequently, a full-connected layer transforms the long vector to a short one to be the output of the cross module, which is represented as
(5)$$CrossNet\left(X_{cross}\right)=Linear\left(Cat\left[{\mathrm{HP}}_{1}^{\left(k_{1}\right)},\cdots ,{\mathrm{HP}}_{i}^{\left(k_{i}\right)},\cdots ,{\mathrm{HP}}_{n}^{\left(k_{n}\right)}\right]\right),$$
where *n* is the number of gated units.

### 3.3. Deep Network

Comparing with image in CV and text in NLP, the data in CTR field lack the spatial or temporal relationship. Those modern DNNs models which excel in CV or NLP have not shown any superiority in CTR prediction task. As a result, we choose plain MLP as the deep module of our model, which is represented as
(6)$$MLP\left(X_{deep}\right)=Linear\left(MLPBlock\left(\cdots MLPBlock\left(X_{deep}\right)\right)\right),$$
MLPBlock(x)=Dropout(relu(BN(Linear(x)))).

### 3.4. Modules Combination

The cross module and deep module are used to learn explicit and implicit interactions, respectively, and both of them are informative to the prediction. The outputs of the two modules are concatenated as a vector, and then fed into the prediction layer, which is represented as
(7)$$\hat{y}=Sigmoid\left(Linear\left(Cat\left[CrossNet\left(X_{cross}\right),MLP\left(X_{deep}\right)\right]\right)\right).$$

CTR prediction task is a binary classification problem, which can be trained with binary cross entropy (Logloss). The loss function of out model is represented as
(8)$$L\left(y,\hat{y}\right)=\frac{1}{N}\sum \left(-y\cdot log\left(\hat{y}\right)-\left(1-y\right)\cdot log\left(1-\hat{y}\right)\right)+\lambda \left\|\Theta \right\|,$$
where *y* is the ground truth, *N* denotes the size of training set, λ denotes the regularization term and ‖Θ‖ denotes the set of parameters in the model.

## 4. Model Analysis

### 4.1. Power Exponent of Feature

The Hadamard product, which calculates the production of all elements in the same position, can be generalized to the power product, then Equation (Equation 4) can be modified as
(9)$${{\mathrm{HP}}^{\prime }}^{\left(k\right)}=V_{t_{l}}⊗V_{t_{2}}⊗\cdots ⊗V_{t_{k}}=\left[\prod_{i=1}^{k}v_{i,1}^{w_{i}},\prod_{i=1}^{k}v_{i,2}^{w_{i}},\cdots ,\prod_{i=1}^{k}v_{i,d}^{w_{i}}\right],$$
where *w* represents the power exponent of the feature. Each feature has its own *w*, and all elements of a feature vector share the same *w*.

AFN+ considers the power exponent of a feature vector as a scaling operation. Hence, *w* is regarded as a scaling factor and can be any value, even a decimal or negative number. Although it seems that introducing power enhances the model’s expressiveness, the power operation requires all inputs to be positive, which severely restricts the embedding representation of the features. The latent space of the embedding vector is compressed to $1/d^{2}$ of its original size, which will weaken the information carried by the embedding vectors and result in a downgrading of the model performance.

Our proposed model simplifies Equation (Equation 9) by restricting the range of the power exponent to {0,1}. The value 1 indicates that the feature will be multiplied, while 0 indicates the opposite. The simplification makes the model of great interpretability.

### 4.2. Complexity Analysis of Cross Network

Our proposed model is mainly inspired by AFN+; hence, we take AFN+ as the comparison target.

**Time Complexity.** Recall that we use *m* to denote the number of feature fields, *d* to denote the dimension of the embedding, *k* to denote the number of the selected features in the gated unit and *n* to denote the number of gated units. The overall time complexity of the cross module in GAIN is O(nkd+nm), while that of AFN+ is O(lmd), where *l* denotes the number of logarithmic neurons. Because n≪l and k≪m, it makes GAIN train faster than AFN+.

**Space Complexity.** For the ease of comparison, the parameters for the batch norm and bias are omitted. The total number of parameters in the cross module of GAIN is nm+nd, while that of AFN+ is $lm+ldh+\left(h-1\right)t^{2}+t$, where *h* and *t* denote the number of hidden layers and the neurons in each layer, respectively. Because n≪l and the MLP in AFN+ is replaced by a single linear layer in GAIN, the space complexity of the cross module in GAIN is far less than in AFN+.

## 5. Experiments

Extensive experiments on two real-world datasets are conducted in this section to answer the following research questions:

**RQ1:** How does GAIN perform compared with those state-of-the-art models?

**RQ2:** How do hyperparameters influence the model performance?

**RQ3:** Can GAIN learn useful feature interactions?

### 5.1. Experiment Setup

Zhu et al. [1] propose an open benchmarking framework for CTR prediction models, which adopts the same data partition, preprocessing and evaluation protocols to make models compete in a fair environment. It is worth noting that, after elaborate tuning, all models perform better than their reported result. Our proposed GAIN is also implemented under this framework.

**Datasets.** Comparative experiments are conducted on two public datasets, Criteo and Avazu, which both come from the Kaggle competition and are widely used in the CTR prediction task. Criteo consists of 39 features, including 26 categorical features and 13 numerical features, 45.8 million samples in total, and the proportion of positive samples is about 25.6%. Avazu consists of 22 categorical features, 36 million samples in total, and the proportion of positive samples is about 17%. The hyperparameters tuning experiments are conducted on Frappe, which consists of 10 categorical features, 288 thousand samples in total, and the proportion of positive samples is about 33.4%.

**Baseline models**. To make a fair comparison, we choose those models which perform best under the benchmarking framework as the baseline. Each model has been fine-tuned and outperforms its reported result by a large margin.

**Wide & Deep** [3]. Wide & Deep assembles two components in parallel. The wide component is composed of a linear combination of raw features and manually designed features. The deep component is implemented by a plain MLP.**IPNN** [5]. IPNN assembles two components in serial. The inner product of the feature embedding vectors are fed to the DNN.**DeepFM** [30]. DeepFM replaces the wide component of Wide & Deep with FM which can automatically learn $2_{rd}$-order interactions.**DCN** [17]. DCN stacks multiple cross layers to learn high-order interactions. Due to its high computational complexity, DCN can hardly learn higher interactions than the $2_{nd}$ order.**xDeepFM** [31]. xDeepFM improves the DCN to learn truly high-order interactions.**AFN+** [32]. AFN+ exploits numerous logarithmic neurons to adaptively learn different high-order interactions.

Although the ONN [18] is demonstrated to significantly improve the prediction accuracy, its high time complexity and memory occupation make it hardly train in our GPU (a single GeForce RTX 3090 with 24G ram). Hence, this model is excluded from our experiments.

**Data preprocessing.** Criteo and Avazu are randomly split into 8:1:1 as the training set, validation set and test set, respectively, while Frappe is split into 7:2:1. In Criteo, the numerical features are discretized to the categorical features, applying log transformation which transforms *x* to $\left\lfloor log^{2}\left(x\right)\right\rfloor$ if x>2. The minimum count of features is set to 10, i.e., infrequent features which appear less than 10 times are replaced with “OOV”. In Avazu, the id field is dropped, and the timestamp field is transformed into three features: hour, weekday and weekend. The minimum count is set to 2.

**Evaluation metrics.** Two metrics, AUC and Logloss, which were widely used in the binary classification task, are adopted to evaluate the model performance. Adam is chosen to be the optimization algorithm, and the learning rate is initialized to 0.001.

**Implementation details.** With respect to the memory limitation of the GPU, the embedding dimension is set to 16, and the mini-batch size is set to 5000 for Criteo and 10,000 for Avazu. All baseline models are fine-tuned with specific hyperparameters for a fair comparison. More details can be found in our GitHub (https://github.com/YaoxunLiu/GAIN, accessed on 18 July 2022).

### 5.2. Performance Comparison (RQ1)

The comparison between GAIN and the baseline models is shown in Table 1. In addition to the Logloss and AUC, we also introduce two metrics, the FLOPs and number of parameters to, respectively, represent the complexity and size of the models. Each model is independently run five times on each dataset with the random seed varying from 2018 to 2022, and the average performance (with their standard deviation in parentheses) is shown in the two leftmost columns. Because the parameters in the embedding layer occupy the vast majority of the parameters in each model, and all the models use the same embedding method, we exclude the embedding layer when counting the parameters.

**Model Effectiveness.** The experimental results show that our proposed GAIN outperforms all baselines on Criteo. Concretely, comparing with the best baseline model DeepFM, GAIN with the hybrid strategy increases the AUC by 0.049% and decreases the Logloss by 0.069%. On Avazu, GAIN achieves comparable results to the best baseline xDeepFM. Although GAIN does not outperform its competitors by a large margin, its model complexity is much lower. Comparing with the best performing baseline model on Criteo, the FLOPs of GAIN on Criteo are 42.3% of that of DeepFM, and the number of parameters of GAIN is only 5.9% of that of DeepFM. A similar situation also occurs in Avazu.

Three strategies for GAIN all behave well on both datasets. The threshold strategy can be easily applied on different datasets because it only relies on one hyperparameter, the threshold ε; however, the upper bound of the model performance is limited because the order distribution generated by the threshold cannot guarantee to contain the optimal schema. Theoretically, the top-*k* strategy can outperform other strategies, yet the huge search space brought by countless combinations of units and orders makes it difficult to find out the optimal schema. By combining the two strategies, the hybrid strategy eliminates their drawbacks and performs best on both datasets. Comparing with the best performing baseline models, the standard deviation of the performance of GAIN is larger. We attribute it to the weight vector in the gated units of GAIN. At the beginning of training, the weights of all features are randomly initialized in each unit, which will affect the selection of the features. Therefore, a different random seed may result in different feature interactions and various model performances. In future work, we will improve our model to reduce the effect of the random seed.

**Model Efficiency.** Efficiency is as important as effectiveness when considering the actual deployment of a model. The memory occupation and time consumption of different models in the training phase are shown in Table 2, and we can observe that the memory occupation of most models is not very different, and GAIN runs faster than the other models. Comparing with the training time, the inference time is more important, especially when the model is asked to respond to users’ requests with low latency. We devise an experiment to compare the training and inference time among different models. For a fair comparison, we set the number of batches to 100 and the batch size to 2000. The experiments for the training and inference time are conducted separately and independently, and the GPU will be warmed-up previously. The results are shown in Figure 3. Apparently, GAIN runs faster both in the training and inference phase. In order to verify GAIN’s efficiency in an online recommendation task, we set the batch size to 1 and perform inference operations on the GPU or CPU alone. Each experiment is conducted 300 times to calculate the average inference time. We also conduct the same experiments on other models. The results in Figure 4 demonstrate the efficiency of GAIN.

### 5.3. Hyperparameter Investigation (RQ2)

In this section, we investigate the effectiveness of three hyperparameters. All the experiments are conducted on Frappe because the time consumption per epoch of the model on this dataset is much less than that of Criteo or Avazu due to its mini size, while the effectiveness of the hyperparameters is similar.

**Threshold ε.** The threshold ε is the most important hyperparameter of the threshold strategy because it determines the distribution of orders. In addition, λ in Equation (Equation 2) also plays an important part. The influences of ε and λ are investigated through a Monte Carlo simulation and shown in Figure 5. Figure 5a provides the relationship curve of ε and the mean of orders in different λ. Apparently, the mean of orders decreases with the increasing ε. The curve of red and blue represents λ=0.1 and λ=1, respectively. We set λ to 0.1 for two reasons. First, the steep curve of λ=0.1 compresses the threshold ε to a smaller range. Second, the relationship between the mean of orders and ε is almost linear in the condition of λ=0.1. The experiments prove that setting λ to 0.1 can narrow the search space of ε, and hence significantly improve the searching efficiency. We simulate the effect of different ε on the order distribution of 100 gated units, and the result of the simulation is shown in Figure 3b, from which we have two observations. First, the orders follow the Gumbel distribution. Second, the orders’ distribution becomes more concentrated when ε increases. For Frappe, which consists of 10 features, useful high-order interactions are mostly concentrated in the range [3,7]. Therefore, we set the mean of orders to 5 and preset the value range of ε to [0.0943,0.0963] according to the red curve in Figure 5a. After enumerating all possible values in the range, we finally find out the best threshold ε=0.0957.

**Number of gated units *n*.** Each gated unit can adaptively learn an arbitrary-order interaction. However, more units cannot always bring a better performance. Extensive experiments are conducted with a different number of units *n* ranging from 1 to 1000, and the result is shown in Figure 6, in which the fold line of red and blue represents the AUC and Logloss, respectively. We can observe that the model performance fluctuates severely when n≤80, then gradually stabilizes when 80<n≤500 and finally decays when n>500. Considering more units bring a high computational complexity to the model, we set *n* to 40 for Frappe to make a trade-off between complexity and expressiveness.

**Order *k*.** The top-*k* strategy requires an allocation of gated units. Because it is an allocation problem, countless possible combinations make it difficult to find out the best distribution scheme. To simplify this problem, firstly, we only take a single order into consideration. We conduct experiments on all possible mappings of *k* and *n* and consider the values of the Logloss and AUC as 2D coordinates, which are plotted in Figure 5. We use different sizes of dots to represent different numbers of units and use different colors to represent different *k*. The x-axis and y-axis represent the Logloss and AUC, respectively. It is worth noting that the x-axis is in a reverse direction; hence, the dots closer to the upper right corner indicate a better performance.

Besides simply allocating all units to a specific *k* order, we can also make a combination of different *k*. In order to restrict the total number of units to 40, we choose small dots from the upper right zone to form a new distribution scheme, e.g., {‘2’:10, ‘3’:20, ‘4’:10}, which consists of ten $2_{nd}$-order units, twenty $3_{rd}$-order units and ten $4_{th}$-order units. The performance of this scheme is plotted with a red star. From Figure 7, we observe that although simple schemes such as {‘6’:40} or {‘5’:40} bring decent results, a deliberate scheme can significantly improve the performance.

### 5.4. The Effectiveness of Feature Interaction Learned by GAIN (RQ3)

The feature combinations learned by gated units are saved in the form of a “hard” vector in which the selected features are represented by the value 1. The application of a “hard” vector not only brings good interpretability but also enables the learned feature interactions to be easily transferred to other models. In order to verify the effectiveness of learned feature interactions, we transfer these “hard” vectors to conventional CTR models, such as LR and FM. These learned interactions are regarded as manually designed features and high-order interactions in LR and FM, respectively. Experiments are conducted on the two original models and their enhanced versions, and the performance comparison is shown in Table 3. We can observe that feature interactions learned by GAIN can help to significantly improve the performance of LR and FM on both of the datasets. Due to the constraints of the theory and the structure of the model itself, the enhanced model cannot make the performance reach the SOTA level; however, the improved performance still demonstrates the effectiveness of the learned feature interactions.

## 6. Conclusions and Future Work

In this work, we propose a novel model, the Gated Adaptive feature Interaction Network (GAIN), which can adaptively learn arbitrary-order feature interactions. Our model consists of four parts: (a) an embedding layer transforms raw inputs to dense vectors with equal length; (b) a cross module explicitly learns feature interactions using multiple gated units; (c) a deep module implicitly learns feature interactions using an MLP; and (d) a prediction layer concatenates the outputs of a cross and deep module and makes the prediction. As the core of the model, the cross module is composed of multiple gated units, each of which can independently learn arbitrary-order interactions in a data-driven manner. Extensive experiments were conducted on two public datasets, and the experimental results demonstrate that our model outperforms the baselines with lower model complexity, which verifies the effectiveness and efficiency of our model. In addition, feature interactions learned by our model can be transferred to LR and FM to improve their performance, proving that the feature interactions learned by the model are indeed useful. For future work, we will further optimize the model structure to reduce the impact of randomly initialized weights on model performance. Moreover, we intend to reduce our model’s complexity to deploy it on devices with limited computing resources.

## Figures and Tables

**Figure 2 sensors-22-07280-f002:**
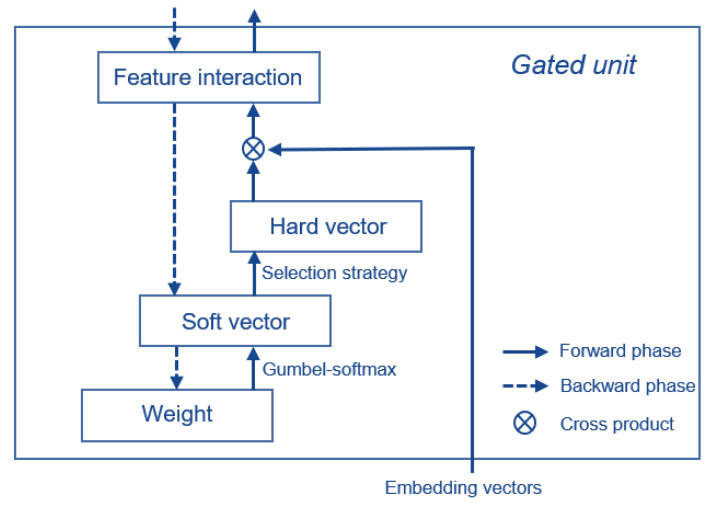
Data flow of a gated unit. Solid line and dashed line denote forward phase and backward phase of training, respectively. At the beginning of training, weights of feature fields are randomly initialized. In the forward phase, “soft” vector is calculated using the trick of Gumbel-softmax, and then “hard” vector is calculated using feature selection strategy. After that, the selected features can participate in interaction. In the backward phase, the gradients are backpropagated step by step, skipping the “hard” vector.

**Figure 3 sensors-22-07280-f003:**
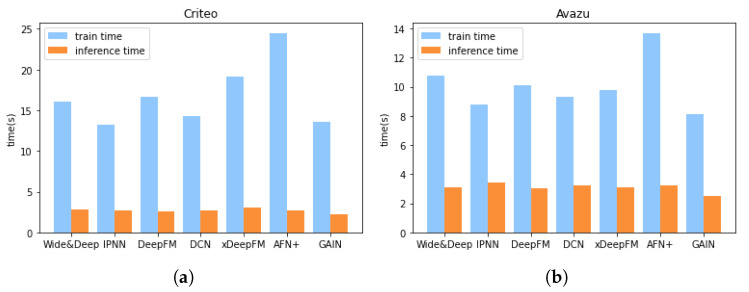
Training and inference time for 100 batches (batch size = 2000) of different models. (**a**) Comparison on Criteo. (**b**) Comparison on Avazu.

**Figure 4 sensors-22-07280-f004:**
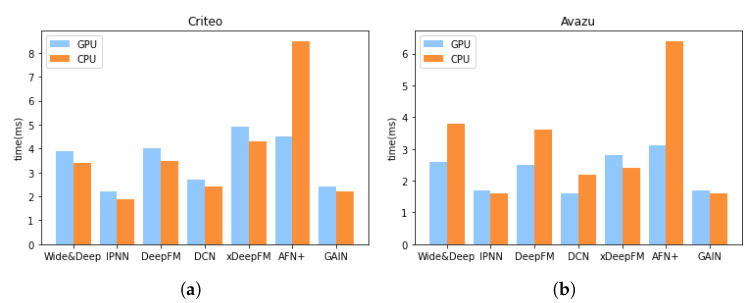
Average inference time of GPU and CPU (batch size = 1) of different models. (**a**) Comparison on Criteo. (**b**) Comparison on Avazu.

**Figure 5 sensors-22-07280-f005:**
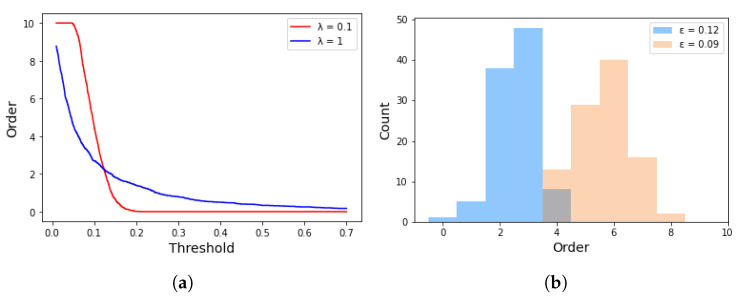
Effects of threshold on orders. (**a**) Relationship between threshold and mean of orders. (**b**) Relationship between threshold and distribution of orders.

**Figure 6 sensors-22-07280-f006:**
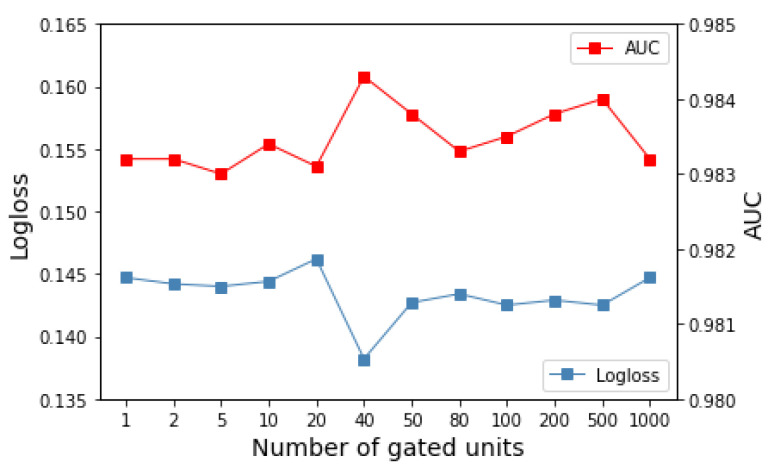
Effects of number of gated units on model performance.

**Figure 7 sensors-22-07280-f007:**
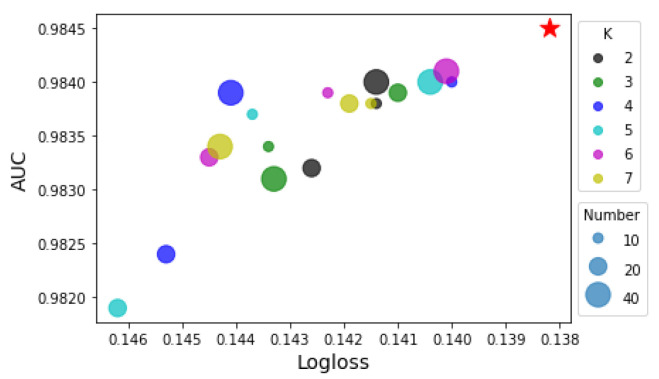
Effects of number of gated units on model performance.

**Table 1 sensors-22-07280-t001:** Performance comparison. The metrics are averaged over 5 independent runs with their standard deviation in parentheses. In the “Parameters” column, parameters from embedding layer are not included.

**Model**	**Criteo**			
	**Logloss**	**AUC**	**FLOPs**	**Parameters**
Wide & Deep	0.4380(3.9E-4)	0.8138(3.2E-4)	4.63 M	32.86 M
IPNN	0.4382(5.1E-4)	0.8137(6.3E-4)	2.39 M	2.36 M
DeepFM	0.4380(6.5E-5)	0.8139(5.5E-5)	4.63 M	32.86 M
DCN	0.4384(2.1E-4)	0.8131(2.6E-4)	4.63 M	4.62 M
xDeepFM	0.4382(9.5E-5)	0.8136(8.4E-5)	11.35 M	33.27 M
AFN+	0.4386(5.6E-4)	0.8130(6.2E-4)	27.72 M	26.88 M
GAIN (top-k)	0.4379(4.8E-4)	0.8140(5.2E-4)	1.96 M	1.95 M
GAIN (threshold)	0.4377(3.0E-4)	0.8142(3.7E-4)	1.96 M	1.95 M
GAIN (hybrid)	0.4376(2.8E-4)	0.8144(1.2E-4)	1.96 M	1.95 M
**Model**	**Avazu**			
	**Logloss**	**AUC**	**FLOPs**	**Parameters**
Wide & Deep	0.3711(1.3E-4)	0.7948(1.3E-4)	12.77 M	21.16 M
IPNN	0.3705(8.6E-5)	0.7955(8.0E-5)	2.67 M	2.67 M
DeepFM	0.3707(2.4E-4)	0.7950(2.8E-4)	12.77 M	21.16 M
DCN	0.3712(2.6E-4)	0.7945(3.8E-4)	8.77 M	8.78 M
xDeepFM	0.3705(6.9E-5)	0.7956(7.3E-5)	3.24 M	9.23 M
AFN+	0.3707(2.0E-4)	0.7955(2.8E-4)	22.16 M	21.63 M
GAIN (top-k)	0.3705(3.2E-4)	0.7956(4.5E-4)	0.75 M	0.75 M
GAIN (threshold)	0.3705(2.4E-4)	0.7955(2.5E-4)	0.75 M	0.75 M
GAIN (hybrid)	0.3704(3.4E-4)	0.7957(3.6E-4)	0.75 M	0.75 M

**Table 2 sensors-22-07280-t002:** Efficiency comparison. The “Memory” columns record GPU’s memory occupation of different models, while the “Time/Epoch” columns record average time consumption per epoch.

Model	Criteo		Avazu	
	Memory	Time/Epoch	Memory	Seconds/Epoch
Wide & Deep	12.08 G	983 s	5.73 G	380 s
IPNN	11.76 G	854 s	5.19 G	371 s
DeepFM	12.07 G	1040 s	4.94 G	394 s
DCN	11.85 G	915 s	5.39 G	375 s
xDeepFM	11.49 G	1388 s	5.14 G	494 s
AFN+	22.96 G	1783 s	10.98 G	612 s
GAIN	13.05 G	881 s	6.66 G	351 s

**Table 3 sensors-22-07280-t003:** Performance of transferring interactions selected by GAIN to LR and FM.

Model	Criteo		Avazu	
	Logloss	AUC	Logloss	AUC
LR	0.4567	0.7934	0.3815	0.7775
FM	0.4431	0.8086	0.3754	0.7887
LR(GAIN)	0.4424	0.8051	0.3801	0.7808
FM(GAIN)	0.4418	0.8093	0.3746	0.7899

## Data Availability

The Criteo dataset supporting this study was obtained from https://www.kaggle.com/c/criteo-display-ad-challenge/data. The Avazu dataset was obtained from https://www.kaggle.com/c/avazu-ctr-prediction/data. The Frappe dataset was obtained from https://www.baltrunas.info/context-aware, accessed on 18 July 2022.

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
