# Peer review of "GAIN: A Gated Adaptive Feature Interaction Network for Click-Through Rate Prediction"

_sensors, 2022, doi:10.3390/s22197280_

Round 1

Reviewer 1 Report

excellent paper, deals with important practical problems of artificial intelligence. I agree with the authors' conclusions, the work should be published as is

Author Response

Dear Editor,

We appreciate you and the reviewers for your precious time in reviewing our paper and providing valuable comments. It was your valuable and insightful comments that led to possible improvements in the current version. The authors have carefully considered the comments and tried our best to address every one of them. We hope the manuscript after careful revisions meet your high standards. The authors welcome further constructive comments if any. 

Below we provide the point-by-point responses. All modifications in the manuscript have been highlighted.

Sincerely,

Yaoxun Liu

Response to Reviewer 1 Comments

Comments: excellent paper, deals with important practical problems of artificial intelligence. I agree with the authors' conclusions, the work should be published as is.

Response: Thank you very much. 

Reviewer 2 Report

1. The abstract should be narrow down on the problem and highlight the need of the proposed work with experimental results.

2. Add the contents in the abstract of the paper and highlight the impact of the proposed work. Result and discussion should be rewritten to summarize the findings/significance of the work.

3. To explore Comparative results with existing approaches/methods relating to the proposed work.

4. The method/approach in the context of the proposed work should be written in detail.

5. In the introduction section, you should give the novelty and the contributions of your works.

6. The literature review is poor in this paper. You must review all significant similar works that have been done. 7.How about the computation complexity of the proposed method?

8. At line 43, LR,FM, Line 56, ONN, …. These abbreviations need to be written in full for the first time.

9. More statistical methods are recommended to analyze the experimental results

10.There are some grammatical mistakes and typo errors. please proof read from native speaker.

Author Response

Dear Editor,

We appreciate you  for your precious time in reviewing our paper and providing valuable comments. It was your valuable and insightful comments that led to possible improvements in the current version. The authors have carefully considered the comments and tried our best to address every one of them. We hope the manuscript after careful revisions meet your high standards. The authors welcome further constructive comments if any. 

Below we provide the point-by-point responses. All modifications in the manuscript have been highlighted. Please see the attachment.

Sincerely,

Yaoxun Liu

Point 1: The abstract should be narrow down on the problem and highlight the need of the proposed work with experimental results.

Response 1: Thanks for your kind reminders. We have rewritten the Abstract. [Pg 1, Ln 2-27]

Point 2: Add the contents in the abstract of the paper and highlight the impact of the proposed work. Result and discussion should be rewritten to summarize the findings/significance of the work.

Response 2: Thanks for your kind reminders. We have rewritten the Conclusion. [Pg 20, Ln 729-755]

Point 3: To explore Comparative results with existing approaches/methods relating to the proposed work.

Response 3: Thanks for your kind reminders. We added experiments to compare our proposed model with baselines in terms of effectiveness, efficiency, model complexity, etc. The comparative results are shown in Table 1, Table 2, Figure 3 and Figure 4. We also analyze the comparative results. [Pg 15, Ln 622-647, Pg 16, Ln 648-662]

Point 4: The method/approach in the context of the proposed work should be written in detail.

Response 4: Thanks for your kind reminders. We have rewritten Section 3.2.1 and Section 3.2.2 to make the description of our method clearer. [Pg 8, Ln 356 - Pg 11, Ln 466]

Point 5: In the introduction section, you should give the novelty and the contributions of your works.

Response 5: Thanks for your kind reminders. We have rewritten the Introduction, the novelty of our work [Pg 3, Ln 123-144], and the contributions. [Pg 4, Ln 146-164]

Point 6: The literature review is poor in this paper. You must review all significant similar works that have been done.

Response 6: Thanks for your kind reminders. We reorganized related works and roughly divided existing models into three categories. Further, we reviewed significant works of the three categories respectively. [Pg 2, Ln 70 – Pg 3, Ln 113]. We also reviewed latest works which are similar to our works, e.g. AutoHash, BH-FIS etc., and added their description to Section 2.3. [Pg 6, Ln 270 – Pg 7, Ln 288]

Point 7: How about the computation complexity of the proposed method?

Response 7: Thanks for your kind reminders. We introduce four metrics to evaluate the computation complexity of our proposed model, i.e., FLOPs, Parameters, Memory and training/inference time. In addition, we devise another two experiments. Firstly, we compare the training and inference time of all models on 100 batches (batch size=100). Secondly, we set batch size to 1, and separately run all models on GPU and CPU alone to compare their inference time. Experimental results demonstrate that the computation complexity of our model is lower than baselines. [Pg 16 – Pg 17]

Point 8: At line 43, LR, FM, Line 56, ONN, …. These abbreviations need to be written in full for the first time.

Response 8: Revised accordingly.

Point 9: More statistical methods are recommended to analyze the experimental results.

Response 9: Thanks for your kind reminders. We supplemented our experiments, and introduced more metrics such as FLOPs, Parameters, training/inference time to validate the performance of our proposed model. We also compared our model with baselines from the perspective of effectiveness and efficiency. [Pg 15, Ln 604 – Pg 17, Ln 662]

Point 10: There are some grammatical mistakes and typo errors. please proof read from native speaker.

Response 10: Thank you very much for the comment. We did our best to correct these mistakes.

Reviewer 3 Report

1. Tables 1 and 2 show that there isn’t much difference in performance between the proposed model and existing state of the art models. The authors should rather emphasize the results of the model analysis.

Author Response

Dear Reviewer,

We appreciate you for your precious time in reviewing our paper and providing valuable comments. It was your valuable and insightful comments that led to possible improvements in the current version. The authors have carefully considered the comments and tried our best to address every one of them. We hope the manuscript after careful revisions meet your high standards. The authors welcome further constructive comments if any. 

Below we provide the point-by-point responses. All modifications in the manuscript have been highlighted. Please see the attachment.

Sincerely,

Yaoxun Liu

Comment: Tables 1 and 2 show that there isn’t much difference in performance between the proposed model and existing state of the art models. The authors should rather emphasize the results of the model analysis.

Response: Thank you very much for the comment. Our work aims to design a lightweight yet effective model. That is to say we are not only concerned with the effectiveness, but also efficiency of model. Our proposed model cannot outperform those baselines by a large margin; however, the computation complexity is lower than its competitors. We supplemented the experiments to compare our model with the baselines from different perspectives. Firstly, we compared the FLOPs and Parameters of all models. Secondly, we compared the training and inference time on 100 batches (batch size=2000). Thirdly, we set batch size to 1, and compared the inference time on GPU and CPU. All experimental results demonstrated that our proposed model is more efficient than the baselines. We also revised the part of Performance Comparison in Section 5.2. [Pg 15, Ln 604 – Pg 17, Ln 662]

Reviewer 4 Report

The author presents a novel feature interaction method by adaptive feature interaction technics. 

Here are some suggestions which I think can make the paper even better:

1. In table 1, the author showed model speed by showing time per epoch. However, it would be great if we could also show the computation cost of Flops or MACs. Also, showing running time in minutes is not precise enough.

2. For metrics that evaluate time and memory, the author should add clarification that the time and memory are evaluated in the inference stage and the batch size; It will also be great if the author evaluates the time and efficiency with batch size equal to 1, and also test on CPU only.

3. For AUC, if the author can provide variance to show the improvement is stable.

Author Response

Dear Reviewer,

We appreciate you for your precious time in reviewing our paper and providing valuable comments. It was your valuable and insightful comments that led to possible improvements in the current version. The authors have carefully considered the comments and tried our best to address every one of them. We hope the manuscript after careful revisions meet your high standards. The authors welcome further constructive comments if any. 

Below we provide the point-by-point responses. All modifications in the manuscript have been highlighted. Please see the attachment.

Sincerely,

Yaoxun Liu

Point 1: In table 1, the author showed model speed by showing time per epoch. However, it would be great if we could also show the computation cost of Flops or MACs. Also, showing running time in minutes is not precise enough.

Response 1: Thanks for your nice reminder. We supplemented our experiments. Firstly, we introduced FLOPs to represent the computation cost which is shown in Table 1. Secondly, we made a new Table 2 to show the running time in seconds.

Point 2: For metrics that evaluate time and memory, the author should add clarification that the time and memory are evaluated in the inference stage and the batch size; It will also be great if the author evaluates the time and efficiency with batch size equal to 1, and also test on CPU only.

Response 2: Thanks for your nice reminder. We conducted experiments as you suggested. Firstly, we compared the training and inference time on 100 batches (batch size=2000). [Fig 3, Pg 17] Secondly, we set batch size to 1, and compared the inference time on GPU and CPU. [Fig 4, Pg 17]

Point 3: For AUC, if the author can provide variance to show the improvement is stable.

Response 3: Thanks for your nice reminder. We change the random seed and run all models independently for 5 times. The average AUC and standard deviation are recorded in Table 1 (we use stddev instead of variance because the variance is too small). We can observe that the AUC of our model is better than the baselines, however, the standard deviation of AUC is higher than IPNN and xDeepFM, which demonstrated that our proposed model is not as stable as expected. We attributed this situation to the randomly initialized weights. Although the feature interactions can be learned in a data-driven manner, the initial weights may affect the selection of features, and result in different performance. [Pg 15, Ln 641-647]

In future work, we will further optimize our model to improve the stability.

Round 2

Reviewer 2 Report

This paper can be accepted now.